# ULF: Unsupervised Labeling Function Correction using Cross-Validation for Weak Supervision

**Anastasiia Sedova**[†⋆] and **Benjamin Roth**[†◇]

[†] Faculty of Computer Science, University of Vienna, Austria
[⋆] UniVie Doctoral School Computer Science, University of Vienna, Austria
[◇] Faculty of Philological and Cultural Studies, University of Vienna, Austria
{anastasiia.sedova, benjamin.roth}@univie.ac.at

## Abstract

A cost-effective alternative to manual data labeling is *weak supervision (WS)*, where data samples are automatically annotated using a predefined set of labeling functions (LFs), rule-based mechanisms that generate artificial labels for the associated classes. In this work, we investigate noise reduction techniques for WS based on the principle of $k$-fold cross-validation. We introduce a new algorithm *ULF* for **U**nsupervised **L**abeling **F**unction correction, which denoises WS data by leveraging models trained on all but some LFs to identify and correct biases specific to the held-out LFs. Specifically, ULF refines the allocation of LFs to classes by re-estimating this assignment on highly reliable cross-validated samples. Evaluation on multiple datasets confirms ULF's effectiveness in enhancing WS learning without the need for manual labeling.[1]

## 1 Introduction

A large part of today's machine learning success rests upon large amounts of annotated training data. However, collecting manual annotation (even in a reduced amount, e.g., for fine-tuning large pre-trained models (Devlin et al., 2019), active learning (Sun and Grishman, 2012), or semi-supervised learning (Kozareva et al., 2008)) is tedious and expensive. An alternative approach is *weak supervision* (WS), where data is labeled in an automated process using one or multiple WS sources such as keywords (Hedderich et al., 2021), knowledge bases (Lin et al., 2016), and heuristics (Varma and Ré, 2018), which are encoded as *labeling functions* (LFs, Ratner et al. 2020). LFs are applied to unlabeled datasets to obtain weak training labels, which are cheap, but often conflicting and error-prone, requiring improvement (see Table 1).

We focus on enhancing the quality of weak labels using *k-fold cross-validation*. Intuitively, by

| | Sample | Matched Labeling Functions | Assigned Label |
|---|---|---|---|
| 1 | **CHECK MY** CHANNEL **OUT PLEASE**. I DO SINGING COVERS | keyword_my, keyword_please, regex_check_out | SPAM |
| 2 | Hello! I'm Marian ... I wanted to play my own pop and pop-rock **songs**. It would mean a lot if you could have a look at **my** channel ... if u like, **subscribe** to it! XOXO THANKS!! | keyword_my, keyword_subscribe, keyword_song, textblob_subjectivity | SPAM/HAM |
| 3 | It looks so real and **my** daughter is a big fan and she likes a lot of your **songs**. | keyword_my, short_songs | SPAM/HAM |
| 4 | Follow me on Twitter @mscalifornia95 | no matches | — |

Table 1: Examples of WS annotation for YouTube dataset (Alberto et al., 2015). (1) is classified as spam as all matched LFs belong to the SPAM class. In (2) and (3), there is a conflict as matched LFs belong to different classes. In (4), no LFs matched; such samples are usually filtered out.

leaving out a portion of the data during training, the model avoids overfitting to errors specific to that part. Hence, *a mismatch between predictions of a model trained on a large portion of the dataset and the labels of the held-out portion can indicate potential noise specific to the held-out portion*. Previous cross-validation-based denoising approaches (Northcutt et al., 2021; Wang et al., 2019b) split the data samples into folds *randomly*; a direct application of these methods to WS data ignores valuable knowledge stemming from the WS process. In our work, we leverage this knowledge by splitting the data based on matching LFs in the samples. The intuition is the following: *a mismatch between predictions of a model trained on a large portion of the LFs and labels generated by held-out LFs can indicate noise specific to the held-out LFs*. By performing cross-validation for each LF, noise associated with all LFs can be identified.

In contrast to correcting labels as in previous work for supervised settings, we utilize the cross-validation principle in a WS setting to adjust and improve the *LF-to-class assignment*. In some cases, an LF correctly captures some samples but mislabels others. For example, one of the LFs used to annotate the YouTube dataset (Alberto et al., 2015) is the keyword *my,* which is effective in identifying spam messages such as *subscribe to my chan-*

---

[1]We make our code available within the Knodle framework (Sedova et al., 2021): https://github.com/knodle.

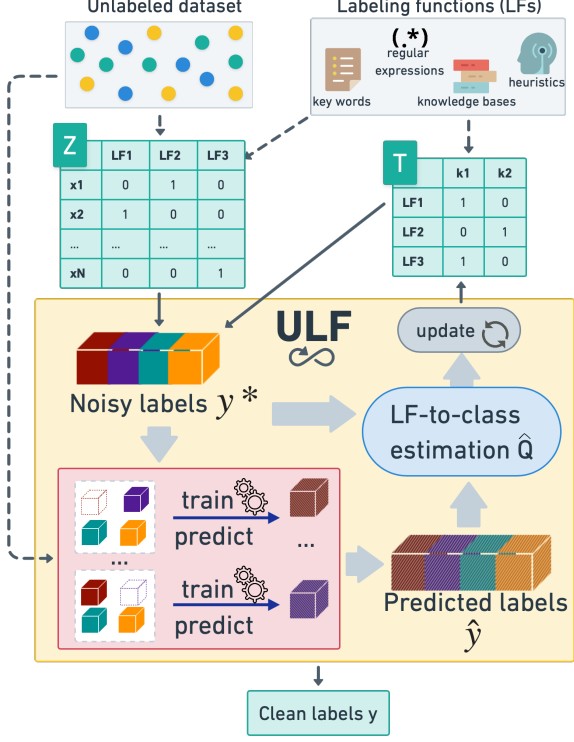

Figure 1: ULF. Noisy training labels $y*$ are obtained by multiplying the matrices $Z$ and $T$. The most confident predictions $\hat{y}$ are calculated using $k$-fold cross-validation and used to estimate new LFs-to-class correspondence and update the $T$ matrix. The clean labels are obtained by multiplying the updated $T$ and $Z$ matrices.

*nel* or *check my channel out* (see Table 1). However, considering this LF as solely indicative of the spam class would be unwarranted, as numerous non-spam messages also contain the word *my* (e.g., Sample 3). The weak labeling of such samples can results in a tie (i.e., one vote for SPAM and one vote for HAM), potentially leading to incorrect label assignments through (random) tie-breaking.

To address such cases, we introduce a new method ULF: **U**nsupervised **L**abeling **F**unction correction (Figure 1), which comprises both erroneous labels detection and correction by leveraging weakly supervised knowledge. ULF re-estimates the joint distribution between LFs and class labels during cross-validation based on highly confident class predictions and their co-occurrence with matching LFs. Importantly, this reestimation is performed *out-of-sample*, meaning it is guided by the data itself without involving additional manual supervision. Instead of a hard assignment of naive WS (i.e., an LF either corresponds to the class or not), ULF performs a fine-adjusted one, which helps to correct the label mistakes. For ex-

ample, such assignment reduces the association of the LF *"my"* with the SPAM class, resulting in dominant HAM probability in Samples 2 and 3 in Figure 1. Moreover, ULF successfully labels samples with no LFs matched, as, e.g., Sample 4, unlike other methods that filter them out (Ratner et al., 2020). We conduct extensive experiments using feature-based and pre-trained models to demonstrate the effectiveness of our method. To the best of our knowledge, we are the first to adapt cross-validation denoising methods to WS problems and refine the LFs-to-class allocation in the WS setting.

## 2   Related Work

Weak supervision (WS) has been widely applied to different tasks across various domains, such as text classification (Zeng et al., 2022), relation extraction (Datta and Roberts, 2023), named entity recognition (Wang et al., 2022), video analysis (Chen et al., 2023), medical domain (Fries et al., 2021), image classification (Yue et al., 2022). Weak annotations are easy to obtain but prone to errors. Approaches to improving noisy data include building a specific model architecture (Karamanolakis et al., 2021), using additional expert annotations (Mazzetto et al., 2021), identifying and removing or downweighting harmful samples (Northcutt et al., 2021; Sedova et al., 2023), or learning from manual user guidance (Boecking et al., 2021; Chatterjee et al., 2020). ULF is compatible with any classifier and do not require any manual supervision; instead of removing the samples, ULF *corrects* the labels, utilizing as much WS data as possible. $K$-fold cross-validation, a reliable method for assessing trained model quality (Wong and Yeh, 2019), is also often used to detect errors in manual annotations (Northcutt et al., 2021; Wang et al., 2019b,a; Teljstedt et al., 2015), but has not been applied to a WS setting. We propose WS extensions to some of these methods in Appendix B and use cross-validation in ULF.

## 3   ULF: Unsupervised Labeling Function Correction

In this section, we present the key elements of ULF. More details can be found in Appendix A, the pseudocode is provided in Algorithm 1.

Given a dataset $X = \{x_1, x_2, ..., x_N\}$ to be used for $K$-class classifier training. In WS setting, we do not have any gold training labels, but only a set of LFs $L = \{l_1, l_2, ..., l_L\}$. An LF $l_j$ *matches* a sample $x_i$ if some condition formulated in $l_j$

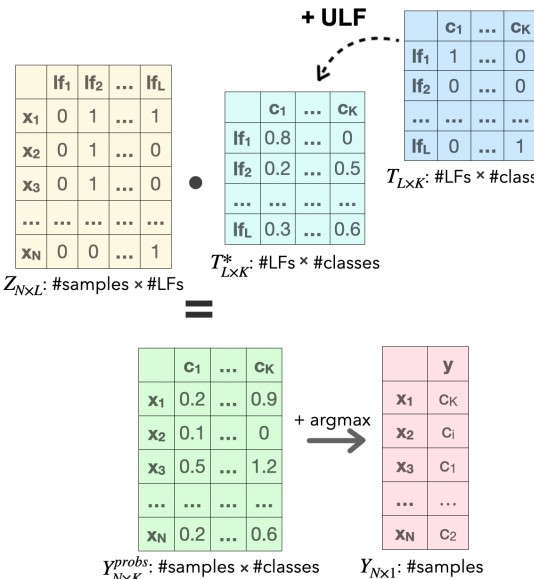

Figure 2: Weak annotation encoded with $Z$ and $T$ matrices, with Z containing LFs matches in samples, and T representing LF-to-class mapping. $T*$ is an improved version of $T$ with ULF. Applying $Z$ and $T$ (or $T*$) matrices multiplication and majority vote yields labels $Y$.

holds for $x_i$. Following Sedova et al. (2021), we store this information in a binary matrix $Z_{N \times L}$, where $Z_{ij}f = 1$ means that LF $l_j$ matches sample $x_i$. A set of LFs matched in sample $x_i$ is denoted by $L_{x_i}$, where $L_{x_i} \subset L$ and $|L_{x_i}| \in [0, |L|]$. The LFs to class correspondence is stored in a binary matrix $T_{L \times K}$, where $T_{ij} = 1$ means the LF $l_i$ corresponds to class $j$ (i.e., $l_i$ *assigns* the samples to the class $j$)[2]. The weak training labels $\tilde{Y} = \{\tilde{y}_1, \tilde{y}_2, ..., \tilde{y}_n\}$, $\tilde{y}_j \in K$ are obtained by multiplying $Z$ and $T$, apply majority vote, and break the ties randomly. The main goal of the ULF algorithm is to refine the $T$ matrix. The graphical explanation is provided in Figure 2.

First, class probabilities are predicted for each sample using $k$-fold cross-validation on the training set $X$ and weak labels $\tilde{Y}$. We propose different ways of splitting the data into $k$ folds $f_1, ..., f_k$, with the most reliable method being **splitting by signatures** (refer to Appendix A for details and other possible splitting methods). The samples' *signatures*, i.e., the sets of LFs matched in each sample, are collected, split into $k$ folds, and used to create data folds: $X_{train_i} = \{x_j | L_{x_j} \notin f_i\}$, $X_{out_i} = \{x_j | L_{x_j} \in f_i\}$ (1). Next, $k$ models are separately trained on each of $k-1$ folds and applied

---

[2]The initial manual class assignment is typically such that each LF corresponds to one class, covering a prototypical case. However, assigning an LF to multiple classes is theoretically possible and compatible with ULF.

to the held-out folds, resulting in *out-of-sample* predicted probabilities $P_{N \times K}$. The *out-of-sample* label $\hat{y}_i$ for each sample $i$ is determined by selecting the class with the highest probability, provided that this highest probability exceeds the *class average threshold* $t_j$:

$$t_j := \frac{\sum_{x_i \in X_{\tilde{y}=j}} p(\tilde{y} = j; x_i, \theta)}{|X_{\tilde{y}=j}|} \quad (2)$$

That is, a sample $x_i$ is *confidently* assigned to class $j$ if the out-of-sample probability of it belonging to class $j$ is higher than the average out-of-sample probability of all samples initially assigned to this class. If no probability exceeds the class thresholds (e.g., all probabilities are equally small), the sample is disregarded as unreliable for further calculations.

These assignments are used to build an LFs-to-classes confidence matrix $C_{L \times K}$, which estimates the joint distribution between *matched LFs* and *predicted labels*. For each LF $l_i$ and each class $k_j$, the confidence matrix $C_{L \times K}$ is populated by counting the number of samples that have LF $l_i$ matched and confidently assigned to class $k_j$:

$$C_{l_i, \hat{y}_j} = |\{x_i \in X : \hat{y}_i = \tilde{y}_j, l_i \in L_{x_i}\}|. \quad (3)$$

---

**Algorithm 1:** ULF: Unsupervised Labeling Function Correction for Weak Supervision

**Input:** unsupervised training data $X$, samples to LFs matrix $Z_{N \times L}$, LFs to classes matrix $T_{L \times K}$, CV model $g(\cdot; \theta_0)$, end model $h(\cdot; \theta_0')$

1   Calculate noisy labels $\tilde{Y} \leftarrow ZT$
2   **for** $iter = 1, 2, ..., I$ **do**
3      Split the data into $k$ folds $f_1, ..., f_k$
4      **for** $f_i, i \in [1, 2, ..., k]$ **do**
5          Build $X_{train_i}, X_{out_i}$ sets (Eq. 1)
6          $\tilde{Y}_{train_i} = \{\tilde{y} \in \tilde{Y} : \forall x \in X_{train_i}\}$
7          $\theta_i = train(X_{train_i}, \tilde{Y}_{train_i})$
8          Calculate $p(\tilde{y} = j; x_i, \theta_i)$ for $\forall x_i \in X_{out_i}, 1 \le j \le K$
9      Calculate labels $\hat{y}_i$ w.r.t. the thresholds $t_j$ (Eq. 2)
10     Calculate LFs-to-class confidence matrix $C_{l, \hat{y}}$ (Eq. 3)
11     Estimate $\hat{Q}_{l, \hat{y}}$ joint matrix (Eq. 4)
12     Recalculate $T^*$ matrix (Eq. 5)
13     Calculate improved labels $\tilde{Y} \leftarrow ZT^*$
14   $\theta' = train(X, \tilde{Y})$
    **Output:** Trained $\theta'$

| | YouTube | Spouse | TREC | SMS | Yorùbá | Hausa | Avg |
|---|---|---|---|---|---|---|---|
| Gold | 98.8 | - | 96.6 | 97.7 | 67.3 | 83.5 | 88.8 |
| Majority Vote (MV) | 93.2 | 21.3 | 68.6 | 93.0 | 48.1 | 43.9 | 61.4 |
| MeTaL (Ratner et al., 2019) | 96.0 | 19.6 | 55.8 | 89.2 | 58.6 | 41.6 | 60.1 |
| Snorkel-DP (Ratner et al., 2020) | 95.6 | 32.6 | 61.8 | 94.6 | 58.7 | 45.7 | 64.8 |
| FlyingSquid (Fu et al., 2020) | 94.0 | 14.9 | 35.8 | 23.7 | 32.4 | 45.1 | 41.0 |
| WeaSEL (Cachay et al., 2021) | 96.0 | 14.9 | 64.4 | 23.6 | 49.6 | 43.2 | 48.6 |
| FABLE (Zhang et al., 2023) | 94.8 | 27.8 | 54.6 | 91.1 | 23.2 | 18.6 | 51.7 |
| MV + Cosine (Yu et al., 2021) | 96.4 | 33.3 | 65.8 | 93.6 | 52.6 | 45.4 | 64.5 |
| MeTaL + Cosine | 95.6 | 26.9 | 67.4 | 80.7 | **62.0** | 45.5 | 63.0 |
| Snorkel-DP + Cosine | 96.0 | 28.1 | 73.8 | 96.1 | 55.0 | 46.5 | 65.9 |
| FlyingSquid + Cosine | 95.6 | 24.9 | 38.6 | 90.1 | 33.3 | 41.5 | 54.0 |
| FABLE + Cosine | 94.0 | 33.9 | 70.6 | **97.7** | 60.1 | 44.7 | 66.8 |
| **ULF (Ours)** | **96.8** | **36.9** | **76.8** | 96.2 | 55.8 | **48.2** | **68.4** |

Table 2: ULF experimental results with pre-trained language models. Accuracy is reported for YouTube and TREC, F1 score is presented for other datasets to account for class imbalance. The Gold baseline is not applied to the Spouse dataset due to the absence of gold labels. Hyper-parameters were obtained through a random search with 10 initialization for energy considerations (not a grid search, as in, e.g., Zhang et al. 2021).

Subsequently, $C_{L \times K}$ is calibrated and normalized to $\hat{Q}_{L \times K}$ in order to align with the data proportions of the $Z$ matrix:

$$\hat{Q}_{l_i, \hat{y}_j} = \left( C_{l_i, \hat{y}_j} \cdot \sum_{m=1}^{L} Z_{l_m, \hat{y}_j} \right) / \left( \sum_{m=1}^{L} C_{l_m, \hat{y}_j} \right), \quad (4)$$

where $\sum_{\substack{i \in L, \\ j \in K}} C_{l_i, \hat{y}_j} = n$, $\sum_{j=1}^{K} Z_{l_m, \hat{y}_j} = \sum_{j=1}^{K} \hat{Q}_{l_m, \hat{y}_j}$. This calibration ensures that $\hat{Q}_{L \times K}$ sums up to the total number of training samples, and the sum of counts for each LF is the same as in the original $Z$ matrix; thus, $\hat{Q}_{L \times K}$ can be utilized as a cross-validated re-estimation of $T$. Finally, a refined $T^*$ matrix is calculated as follows:

$$T^* = p * \hat{Q} + (1 - p) * T. \quad (5)$$

Here, the hyperparameter $p$, $p \in [0, 1]$, determines the extent to which information from the original $T$ matrix should be retained. The resulting $T^*$ matrix is utilized to generate improved labels for additional ULF iterations or training the final classifier.

**Unlabeled samples.** ULF also takes advantage of unlabeled samples that do not have any LFs matched. A portion of these samples, determined by the hyperparameter $\lambda$, is randomly labeled and included in cross-validation training, with reestimation in subsequent iterations. To leverage *all* unlabeled samples in a fine-tuning-based setting, we also include the *optional* Cosine self-training step (Yu et al., 2021), which can be executed during cross-validation and/or final classifier training.

| CV / Final | YouTube | Spouse | TREC | SMS | Yorùbá | Hausa |
|---|---|---|---|---|---|---|
| FT_FT | 96.8 | 22.0 | 68.2 | 96.1 | 54.6 | 43.0 |
| FT_Cos | 94.4 | 36.9 | 76.6 | 96.2 | 54.2 | 48.2 |
| Cos_FT | 95.2 | 21.3 | 68.6 | 96.1 | 55.8 | 43.6 |
| Cos_Cos | 94.8 | 33.0 | 76.8 | 96.1 | 54.2 | 44.5 |

Table 3: Results of all ULF combinations of cross-validation and final model (simple fine-tuning (FT) or followed by additional Cosine (Cos) training).

## 4 Experiments

**Datasets and baselines.** We evaluate ULF on four WS English datasets: (1) YouTube Spam Classification (Alberto et al., 2015); (2) Spouse Relation Classification (Corney et al., 2016); (3) Question Classification from TREC-6 (Li and Roth, 2002); (4) SMS Spam Classification (Almeida et al., 2011), and two topic classification WS African datasets: (5) Yorùbá and (6) Hausa (Hedderich et al., 2020). For all datasets, we utilize the LFs provided by dataset authors.

We compare our results towards the (1) *Gold* baseline (the only classifier which exploits gold labels) and the most popular and recent WS baselines: (2) *Majority Vote*, (3) *MeTaL* (Ratner et al., 2019), (4) *Snorkel-DP* (Ratner et al., 2020), (5) *Flying Squid* (Fu et al., 2020), (6) *WeaSEL* (Cachay et al., 2021), and (7) *FABLE* (Zhang et al., 2023). More details are provided in Appendices C and D.

**Results.** We run experiments using RoBERTa following Zhang et al. (2021) for English datasets and mulitlingual BERT following Devlin et al. (2019) for others (more implementation details are provided in Appendix F). Table 2 presents the results of the best combination of cross-validation and final models; each of them can be either simple

fine-tuning or followed by Cosine training step (Yu et al., 2021). On average, ULF outperforms the baselines and achieves better results on four out of six datasets. The weakest performance was observed on the Yorùbá dataset, which is explained by the extremely high number of labeling functions (19897) and the smallest training dataset size (1340 samples) when compared to the other datasets.

Results of other combinations are provided in Table 3. Two out of the four combinations achieve average scores of 67.6 and 67.1, demonstrating a better performance compared to the baselines. Although the Cosine contrastive self-training considerably improves the results, the ULF high performance does not rely solely dependent on it. This is evident in the fact that the most effective configuration for all other datasets except TREC incorporates the use of Cosine in only one of the two model training steps. Moreover, FT_FT setting, which does not involve Cosine at all, also demonstrates compatible results across all datasets.

**Case Study.** We provide a YouTube dataset case study. Figure 3 shows the initial and adjusted $T$ matrices after two ULF iterations in FT_FT setting. Some LFs underwent minimal adjustments (such as `keyword_subscribe` and `regex_check_out`, which clearly corresponded to one class), while contentious LFs (like `short_comment`, i.e., short comments are non-spam) were significantly adjusted. The adjustment was slightly improved after the second iteration of ULF; however, a single iteration was already sufficient for most of the settings, as demonstrated by our experiments (see Appendix F). Table 4 shows mislabeled samples and their corrected labels after ULF application. In (1), the original equal voting was changed to 1.58 for HAM and 2.42 for SPAM after $T$ matrix correction, explicitly determining the label as SPAM. Similarly, labels assigned by a clear majority vote, such as (2), were also corrected. Next, there are sam-

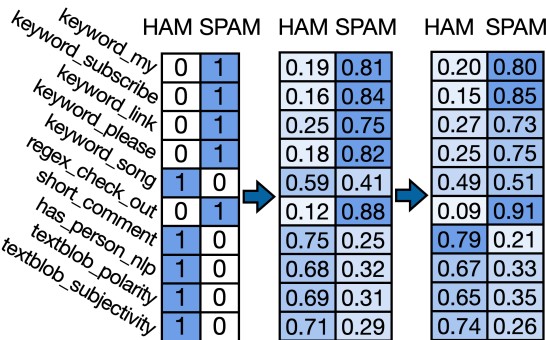

Figure 3: Transformation of $T$ matrix with ULF after first and second iterations.

ples where improved and gold labels do not match (i.e., where ULF, strictly speaking, failed). However, these samples are quite controversial: e.g., (3) might be a spam message if the link was different (our model does not check the link's content), while (4) can be interpreted differently and perceived as a spam comment. Finally, (5), not covered by any LFs and initially randomly assigned to the HAM class, has been corrected to the SPAM class.

## 5 Conclusion & Future Work

In our work, we focused on denoising WS data by leveraging information from LFs. Our approach assumes that the noise specific to some LFs can be detected by training a model that does not use those LFs signals and then comparing its predictions to the labels generated by the held-out LFs. This idea is used in our method ULF, which improves the weak labels based on the data itself, without leveraging external knowledge. Extensive experiments validate the effectiveness of our approach and support our initial hypothesis of the significant role of LFs in denoising WS data. In future work, we plan to try ULF for other tasks, such as sequence tagging and image classification, and perform more experiments on weakly supervised datasets with different peculiarities and in different languages.

| | Sample | LFs matched | Noisy Label | Corrected Label | Gold Label |
|---|---|---|---|---|---|
| 1 | Hello! I'm Marian ... I wanted to play my own pop and pop-rock **songs**. It would mean a lot if you could have a look at **my** channel ... if u like, **subscribe** to it! XOXO THANKS!! | keyword_my keyword_subscribe keyword_song textblob_subjectivity | HAM | SPAM | SPAM |
| 2 | Nice **song** .See **my** new track. | keyword_my keyword_song textblob_subjectivity | HAM | SPAM | SPAM |
| 3 | 'HAPPY BIRTHDAY KATY :) **http://giphy.com/gifs/birthday-flowers-happy-gw3JY2uqiaXKaQXS/fullscreen** ... | keyword_link textblob_subjectivity | HAM | SPAM | HAM |
| 4 | The little PSY is suffering Brain Tumor and only has 6 more months to live. **Please** pray to him and the best lucks. | keyword_please textblob_subjectivity | HAM | SPAM | HAM |
| 5 | Follow me on Twitter @mscalifornia95 | — | HAM | SPAM | SPAM |

Table 4: Examples of label changes in the YouTube dataset after applying ULF.

## Limitations

In our work, we did not focus on the task of creating labeling functions. Rather, our primary objective is to improve the model performance with a fixed set of already provided labeling functions, and to enable better generalization to new data.

All the datasets and their corresponding labeling functions used in our experiments are weakly supervised datasets that have been extensively utilized in previous research. The provided labeling functions for these datasets, as well as other well-known weakly supervised datasets, are considered reliable. ULF does not require the majority of LFs to have high precision; however, if we consider a significantly different setting where the *majority* of labeling functions are *highly* unreliable (e.g., generated by a noisy automatic process), cross-validation as done in ULF may not be as effective as in a more standard WS setting.

In our experiments, we restricted ourselves to NLP datasets and tasks, as creating labeling functions for weak supervision is particularly intuitive for language-related tasks. We leave the exploration of other data modalities for future research.

## Ethics Statement

While our method can lead to better and more helpful predictions by the models trained on the noisy data we cannot guarantee that these predictions are perfect and can be trusted as the sole basis for decision-making, especially in life-critical applications (e.g. healthcare). Machine learning systems can pick up and perpetuate biases in the data, and if our algorithms are used for real-world applications, the underlying data and the predictions of the resulting models should be critically analyzed with respect to such biases. We build our work on previously published datasets and did not hire annotators.

## Acknowledgement

We thank the anonymous reviewers for their constructive feedback. This research has been funded by the Vienna Science and Technology Fund (WWTF)[10.47379/VRG19008] "Knowledge-infused Deep Learning for Natural Language Processing".

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

# A    Details on ULF method

In this section, we give a more formal description of the ULF algorithm as well as discuss its details.

## A.1    A detailed explanation of Z, T, and Y matrices

Figure 2 represents the weak annotation encoded with $Z$ and $T$ matrices. The matrix $Z_{N \times L}$ represents the information regarding the matches of labeling functions (LFs) in samples. For instance, in the case of keyword-based LFs, an LF matches a sample if this keyword is present in this sample (this sample is then assigned to the class associated with this LF). This matrix is binary: if an LF $l_j$ matches a sample $x_i$, $Z_{ij} = 0$.

The matrix $T_{L \times K}$ signifies the correspondence between LFs and classes. Original $T$ matrix is binary: each element $T_{kl}$ represents whether an LF $l_k$ corresponds to class $l$. $T_{kl} = 1$ indicates that an LF $l_k$ corresponds to class $l$ and assigns this class to all samples where it matches. For example, the keyword *subscribe* corresponds to the class SPAM; any sample containing the word *subscribe* will receive one vote for the SPAM class.

By multiplying the matrices $Z$ and $T$ and applying majority vote, we obtain weak labels $\tilde{Y}$. The $T^*$ matrix denotes the improved version of the $T$ matrix achieved through ULF. Note that the $T^*$ matrix is not binary anymore: instead of hard assignments, it contains soft ones. The improved, clean labels can be obtained by multiplying the improved matrix $T^*$ with the original matrix $Z$.

## A.2    Data Splitting into Folds for Cross-Validation

First, the training data is split into $k$ folds for cross-validation training. We analyzed three possible ways of splitting:

- **randomly (ULF$_{rndm}$)**: the samples are assigned to folds the same way as it would be done in standard $k$-fold cross-validation irrespective of LFs matching;

- **by LF (ULF$_{lfs}$)**: the LFs are randomly split into $k$ folds $\{f_1, ..., f_k\}$ and each fold $f_i$ is iteratively taken as held-out LFs, while others become training LFs. All samples where training LFs match become training set $X_{train_i}$; the rest build a hold-out set and are used for re-estimation $X_{out_i}$:

$$X_{train_i} = \{x_j | L_{x_j} \cap f_i = \emptyset\}$$
$$X_{out_i} = X \setminus X_{train_i} \quad (7)$$

- **by signatures (ULF$_{sgn}$):** for each training sample $x_i$, we define its signature $L_{x_i}$ as the set of matching LFs. For instance, if LFs $lf_1, lf_4,$ and $lf_7$ match in a sample $x_i$, its signature is represented as $\{1, 4, 7\}$. Next, the signatures are split into $k$ folds, each of which becomes in turn a test fold, while others compose training folds. All samples with signatures present in the training folds are considered as training samples $X_{train_i}$, while the remaining samples form the hold-out set $X_{out_i}$.

$$X_{train_i} = \{x_j | L_{x_j} \notin f_i\}$$
$$X_{out_i} = \{x_j | L_{x_j} \in f_i\} \quad (8)$$

After the data is split into folds, $k$ models are separately trained on $X_{train_i}, i \in [1, k]$ and applied on the held-out folds $X_{out_i}$ to obtain the out-of-sample predicted probabilities $P_{N \times K}$.

### A.3 Out-of-sample Labels Calculation

The predicted probabilities $P_{N \times K}$ are used to calculate the out-of-sample labels $\hat{y}$.

The class with the highest probability is selected:

$$\hat{y}_i = \arg\max_{1 \leq j \leq K} p(\tilde{y} = j; x_i, \theta), \quad (9)$$

if and only if the probability $p(\tilde{y} = j; x_i, \theta)$ exceeds the class $j$ average threshold $t_j$:

$$t_j := \sum_{x_i \in X_{\tilde{y}=j}} p(\tilde{y} = j; x_i, \theta)/|X_{\tilde{y}=j}| \quad (10)$$

In other words, the class average threshold $t_j$ is calculated by summing up the probabilities of the $j$ class for samples initially assigned to that class and then dividing it by the number of those samples. A sample $x_i$ is considered as belonging to the class $j$ if and only if this sample *confidently* belongs to the corresponding class. If no probability exceeds the class thresholds for a sample (e.g., it belongs to all classes with equally small probabilities), it is disregarded in further calculations as unreliable.

### A.4 LFs-to-Class Estimation Matrix

To refine the LFs to class allocation, ULF re-estimates the joint distribution between *matched LFs* and *predicted labels*. For each LF $l_i$ and each class $k_j$, the confidently assigned to the class $k_j$ samples with the LF $l_i$ matched are calculated; the counts are saved as a *LFs-confident matrix $C_{L \times K}$*:

Next, the confident matrix is calibrated and normalized to $\hat{Q}_{L \times K}$ to correspond to the data proportion in the $Z$ matrix:

$$\hat{Q}_{l_i, \hat{y}_j} = \left( C_{l_i, \hat{y}_j} \cdot \sum_{m=1}^{L} Z_{l_m, \hat{y}_j} \right) / \left( \sum_{m=1}^{L} C_{l_m, \hat{y}_j} \right), \quad (11)$$

where $\sum_{\substack{i \in L, \\ j \in K}} C_{l_i, \hat{y}_j} = n$, $\sum_{j=1}^{K} Z_{l_m, \hat{y}_j} = \sum_{j=1}^{K} \hat{Q}_{l_m, \hat{y}_j}$.

---

**Algorithm 2:** Train $(X, \tilde{Y})$ in feature-based ULF

**1** $\theta = \text{AdamW}(\theta, X, \tilde{Y})$

**Output:** Trained $\theta$

---

**Algorithm 3:** Train $(X, \tilde{Y})$ in ULF with pretrained language model fine-tuning (optionally: with additional Cosine self-training step)

**1** $X_{lab} = \{x \in X : |L_{x_i}| > 0\}$
**2** $\tilde{Y}_{lab} = \{\tilde{y} \in \tilde{Y} : \forall x \in X_{lab}\}$
   *# 1. fine-tune $\theta$ with $X_{lab}$ and $\tilde{Y}_{lab}$*
**3** $\theta = \text{AdamW}(\theta, X_{lab}, \tilde{Y}_{lab})$
   *# 2. (optional) contrastive self-training of $\theta$ with $X$*
**4** Calculate pseudo labels $y_{\text{psd}}$
**5** **for** *step = 1, 2, ..., num_steps* **do**
**6** |  Select confident samples
**7** |  Calculate classification loss $L_c(\theta, y_{\text{psd}})$, contrastive regularizer $R_1(\theta, y_{\text{psd}})$, confidence regularizer $R_1(\theta)$ (see Yu et al. (2021) for exact formulas and explanation)
**9** |  $L(\theta, y_{\text{psd}}) = L_c + \lambda R_1 + R_1$
**10** |  $\theta = \text{AdamW}(\theta, X)$
**Output:** Trained $\theta$

---

### A.5 T Matrix Update

The joint matrix $\hat{Q}$ is used for improving the LFs-to-class matrix $T$ that contains the initial LFs-to-class allocations. $T$ and $\hat{Q}$ are summed with multiplying coefficients $p$ and $1-p$, where $p \in [0, 1]$. The value of $p$ balances the initial manual label assignment with the unsupervised re-estimation and determines how much information from the estimated assignment matrix $\hat{Q}$ should be preserved in the refined matrix $T^*$:

$$T^* = p * \hat{Q} + (1 - p) * T \quad (12)$$

With the multiplication of $Z$ and the newly re-calculated $T^*$ matrices, an updated set of labels $Y^*$ is calculated. It can either be used for rerunning the denoising process or training the end classifier.

# B    Weakly Supervised Extension of Denoising Models

To validate our hypothesis regarding the importance of considering labeling functions in noise detection in WS data, we adopt two cross-validation-based methods originally designed for denoising manually annotated data: CrossWeigh (Wang et al., 2019b) and Cleanlab (Northcutt et al., 2021). We adapt these methods for the weakly supervised setting and introduce our extensions: **WSCrossWeigh** and **WSCleanlab**. A key difference between the original methods and our extensions is the approach used to split the data into folds for cross-validation training. Instead of random splitting, we split the data based on the *labeling functions* that match in the samples.

In this section, we outline the original methods and introduce our WS extensions. Additionally, we conducted experiments, which are detailed in Appendix E.

## B.1    Weakly Supervised CrossWeigh

The original CrossWeigh framework (CW, Wang et al. 2019b) was proposed for tracing inconsistent labels in the crowdsourced annotations for the named entity recognition task. After randomly splitting the data into $k$ folds and building $k$ training and hold-out sets, CrossWeigh additionally filters the training samples that include the entities matched in hold-out folds samples. The intuition behind this approach is that if an entity is constantly mislabeled, the model would be misguided, but the model trained without it would eliminate this confusion. We consider this approach quite promising for detecting unreliable LFs in weakly supervised data similarly. If a potentially erroneous LF systematically annotates the samples wrongly, reliable model-trained data without it will not make this mistake in its prediction, and, thus, the error will be traced and reduced. Our new **Weakly Supervised CrossWeigh** method (WSCW) allows splitting the data not entirely randomly but considering the LFs so that all LFs matched in a test fold are eliminated from the training folds. More formally, firstly, we randomly split labeling functions $L$ into $k$ folds $\{f_1, ..., f_k\}$. Then, we iteratively take LFs from

each fold $f_i$ as test LFs and the others as training LFs. So, all samples where no LFs from hold-out fold match become training samples, while the rest are used for testing.

$$X_{out_i} = X \setminus X_{train_i}$$

After that we train the $k$ separate models on $X_{train_i}$ and evaluate them on $X_{out_i}$. In the same way, as in the original CrossWeigh algorithm, the labels predicted by the trained model for the samples in the hold-out set $\hat{y}$ are compared to the initial noisy labels $y$. All samples $X_j$ where $\hat{y}_j \neq y_j$ are claimed to be potentially mislabeled; their influence is reduced in further training. The whole procedure of error detection is performed $t$ times with different partitions to refine the results. The sample weights $w_{x_N}$ are then calculated as $w_{x_j} = \epsilon^{c_j}$, where $c_j$ is the number of times a sample $x_j$ was classified as mislabeled, $0 \leq c_j \leq t$, and $\epsilon$ is a weight reducing coefficient.

## B.2    Weakly Supervised Cleanlab

The second method we introduce is **Weakly Supervised Cleanlab** (WSCL) - an adaptation of Cleanlab framework (Northcutt et al., 2021) for weak supervision. In the same way as in WSCW, not data samples, but the labeling functions $L$ are split into $k$ folds $\{f_1, ..., f_k\}$ and used for building the $X_{train_i}$ and $X_{out_i}$ sets, $1 < i < k$, for training $k$ models. In contrast to WSCW, for each sample, $x_i$ the label is not directly predicted on the $X_{out_i}$, but the probability vector of class distribution $\hat{p}(y = j; x_i, \theta), j \in K$ is considered. The exact labels $\hat{y}$ are calculated later on with respect to the class expected self-confidence value $t_j$ (see Northcutt et al. 2021):

$$t_j := \frac{\sum_{X_j} \hat{p}(y = j; x_i, \theta)}{|X_j|}, \quad (13)$$

where $X_j = \{x_i \in X_{y=j}\}, 1 < j < c$

A sample $x_i$ is considered to confidently belong to class $j \in K$ if the probability of class $j$ is greater than expected self-confidence for this class $t_j$ or the maximal one in the case of several classes is probable:

$$\hat{y}_i = \underset{\substack{j \in [K]: \\ \hat{p}(y=j; x_i, \theta) \geq t_j}}{\arg \max} \hat{p}(y = j; x_i, \theta) \quad (14)$$

The samples with no probability that would exceed the thresholds have no decisive label and do not participate in further denoising.

After that, a class-to-class confident joint matrix $C_{y,\hat{y}}$ is calculated, where:

$$C_{y,\hat{y}}[j][k] = |\{x_i \in X | y_i = j, \hat{y}_i = j\}|$$

Notably, $C_{y,\hat{y}}$ contains only the information about correspondence between noisy and out-of-sample predicted labels (the same way as in North-cutt et al. (2021)). So, it gives the idea about the number of samples with presumably erroneous noisy labels $y$ but does not give us any insights about the erroneous labeling functions that assigned this noisy label to this sample (in contrast to the ULF approach we present in Section 3).

The confident matrix $C_{y,\hat{y}}$ is then calibrated and normalized in order to obtain an estimate matrix of the joint distribution between noisy and out-of-sample predicted labels $\hat{Q}_{y,\hat{y}}$, which determines the number of samples to be pruned. We perform the pruning by noise rate following the Cleanlab default setting: $n \cdot Q_{y_i,\hat{y}_j}, i \neq j$ samples with $max(\hat{p}(y = j) - \hat{p}(y = i))$ are eliminated in further training.

## C  Datasets

In this section, we give a brief overview of the dataset and the examples of labeling function we used in our experiments. The dataset statistics is provided in Table 5.

**YouTube**  (Alberto et al., 2015) A spam detection dataset was collected from the YouTube video comments. The samples that are not relevant to the video (e.g. advertisement of user's channel or ask for subscription) are classified as SPAM, while others belong to the HAM class. We use the same labeling functions as in Ratner et al. (2020); they were created using keywords, regular expressions, and heuristics. For example, a labeling function KEYWORD_MY corresponds to class SPAM, meaning that if a sample contains the word "my" it will be assigned to the SPAM class. Among other labeling functions are KEYWORD_SUBSCRIBE, KEYWORD_PLEASE, KEYWORD_SONG (keyword-based), SHORT_COMMENT (if a comment is short, it is probably not spam; thus, samples less than 5 words long would be classed as HAM).

**Spouse**  (Corney et al., 2016) A relation extraction dataset based on the Signal Media One-Million News Articles Dataset, which main task is to define whether there is a spouse relation in a sample. We use the Snorkel annotation (Snorkel); the labeling functions were created based on keywords (e.g., husband), spouse relationships extracted from DB-Pedia (Lehmann et al., 2014), and language patterns (e.g., check whether the people mentioned in a sample have the same last name).

**TREC**  (Li and Roth, 2002) A question classification dataset that maps each data sample to one of 6 classes. The labeling functions were generated based on keywords (Awasthi et al., 2020), e.g. which, what, located, situated keywords relate a sample to the class LOCATION.

**SMS**  (Almeida et al., 2011) A spam detection dataset comprised of text messages. The annotation proposed by Awasthi et al. (2020) includes keyword-based and regular expression-based labeling functions. For example, a regex-based labeling function:

```
( |^)(won|won)[^\w]* ([^\s]+ )*
(claim,|claim)[^\w]*( |$)
```

corresponds to class SPAM (e.g., as in a sample *449050000301 You have won a ??2,000 price! To claim, call 09050000301.*).

**Yorùbá and Hausa**  (Hedderich et al., 2020) Topic classification datasets of the second (Hausa) and the third (Yorùbá) most spoken languages in Africa comprised of news headlines. The weak keyword rules are provided by the authors.

## D  Baselines

We compare our method ULF towards the most recent weakly-supervised baselines. Note that ULF does not use manually annotated data and cannot be directly compared to models that do (Karamanolakis et al., 2021; Awasthi et al., 2020).

**Gold**  A classifier is trained using the gold, manual labels. It is the only model which is trained with manual labels in our experiments.

**Majority Vote**  A classifier is trained using weak labels obtained by applying labeling functions to the samples, selecting the class with majority voting, breaking the ties randomly.

**MeTaL**  (Ratner et al., 2019) A classifier is trained with labels which are obtained by combining signals from multiple labeling functions and training a hierarchical multi-task network.

|  | **YouTube** | **Spouse** | **TREC** | **SMS** | **Yorùbá** | **Hausa** |
|---|---|---|---|---|---|---|
| Train Data | 1586 | 22254 | 4965 | 4502 | 1340 | 2045 |
| Valid Data | 150 | 2711 | 500 | 500 | 189 | 290 |
| Test Data | 250 | 2701 | 500 | 500 | 379 | 582 |
| #Classes ($K$) | 2 | 2 | 6 | 2 | 7 | 5 |
| #LFs ($L$) | 10 | 9 | 68 | 73 | 19897 | 18624 |
| #Unlabeled | 195 | 16520 | 242 | 2719 | 0 | 0 |
| Avg LF Hits | 1.6 | 33.7 | 1.7 | 0.5 | 3.0 | 2.9 |
| LF Accuracy | $81\% \pm 2.0$ | $53\% \pm 0.6$ | $50\% \pm 2.6$ | $60\% \pm 1.6$ | $55\% \pm 1.5$ | $54\% \pm 0.3$ |
| LF Coverage | 87% | 25% | 85% | 40% | 100% | 100% |

Table 5: Statistics of all the datasets. The LF accuracy metrics are calculated with a majority vote without any model training reported across 10 runs with standard deviation to reduce the instability caused by randomly broken ties.

**Snorkel-DP** (Ratner et al., 2020) A classifier is trained using generative and discriminative Snorkel steps.

**FlyingSquid** (Fu et al., 2020) A classifier is trained using noisy labels that are rectified exploiting an Ising model by a triplet formulation.

**WeaSEL** (Cachay et al., 2021) A classifier is trained using a probabilistic encoder and a downstream model combined with a specifically defined noise-aware loss function. As WeaSEL is an end-to-end system, we were unable to include the Cosine step to it; that is the reason why (potential) WeaSEL+Cosine baseline is absent in Table 2.

**FABLE** (Zhang et al., 2023) A classifier is trained using noisy labels that are inferred leveraging the instance features and the mixture coefficients of the EBCC model.

In all baseline runs, we adhered to the hyper-parameters and their associated search spaces as suggested by the methods' authors. However, for some of the baselines, we have to change the setting proposed in the original papers in order to provide a fair comparison. For instance, in FABLE, we fine-tuned the hyper-parameters of the final classifier training, similar to how we did it for other baselines, even though the authors did not conduct such fine-tuning in their experiments and did not assume the presence of a gold validation set. It is also important to note that due to energy considerations and resource constraints, we retrieved the best hyper-parameter values by random search, rather than grid search. Consequently, for some of the results, direct comparison is not possible.

## E Feature-based Experiments

In Section 4, we presented the results of our fine-tuning-based ULF implementation. In addition to it, we also provide **a feature-based ULF** implementation that does not rely on pre-trained language models but can be run with an arbitrary model for the feature-based prediction.

We run the feature-based ULF experiments for four datasets: YouTube, Spouse, TREC, and SMS. The datasets selection was motivated by previous work that includes feature-based methods (Zhang et al., 2021, 2023) and run experiments with these datasets. This choice allows for a direct comparison of our results with those studies. (For the fine-tuning-based experiments, which typically yield better results, we expanded our experimental setup and included the low-resource language datasets Yorùbá and Hausa, see Section 4.)

We compare the feature-based ULF approach to the same weakly supervised baselines used for comparing the fine-tuning-based ULF approach. Those are: Majority Vote, MeTaL (Ratner et al., 2019), Snorkel-DP (Ratner et al., 2020), FlyingSquid (Fu et al., 2020), FABLE (Zhang et al., 2023), and WeaSEL baseline (Cachay et al., 2021), which also uses logistic regression in the setting presented in the original paper. Additionally, we include two methods for learning with noisy labels we already discussed in Appendix B - CrossWeigh (Wang et al., 2019b) and Cleanlab (Northcutt et al., 2021), also feature-based in the original setting - together with our weakly supervised extensions WSCrossWeigh and WSCleanlab. The feature-based ULF method was run in all possible data splitting settings: randomly ($ULF_{rndm}$), by LFs ($ULF_{lfs}$), and by signatures ($ULF_{sgn}$), see Appendix A for more

|  | **YouTube** (Acc) | **Spouse** (F1) | **TREC** (Acc) | **SMS** (F1) | **Avg** |
|---|---|---|---|---|---|
| Gold | 94.0 ± 0.0 | - | 81.9 ± 0.5 | 91.6 ± 0.2 | 89.3 |
| Majority Vote (MV) | 90.7 ± 1.4 | 24.0 ± 0.1 | 57.6 ± 5.0 | 90.8 ± 1.0 | 65.8 |
| MeTaL* (Ratner et al., 2019) | 89.8 ± 0.8 | 21.8 ± 0.8 | 59.5 ± 1.8 | 89.1 ± 0.9 | 65.1 |
| Snorkel-DP* (Ratner et al., 2020) | 90.0 ± 0.8 | 24.8 ± 1.1 | **64.1** ± 4.4 | 25.4 ± 0.7 | 51.1 |
| FlyingSquid* (Fu et al., 2020) | 87.7 ± 0.8 | 28.7 ± 2.0 | 34.2 ± 2.0 | 66.0 ± 5.5 | 54.1 |
| WeaSEL (Cachay et al., 2021) | 53.0 ± 0.2 | 8.9 ± 2.4 | 27.6 ± 0.1 | 24.0 ± 0.1 | 28.4 |
| FABLE (Zhang et al., 2023) | 91.1 ± 0.2 | 23.6 ± 0.2 | 46.0 ± 0.7 | 64.7 ± 1.1 | 56.3 |
| CrossWeigh (Wang et al., 2019b) | 90.1 ± 0.6 | 41.6 ± 0.5 | 40.1 ± 0.1 | 79.6 ± 1.4 | 62.8 |
| Cleanlab (Northcutt et al., 2021) | 86.9 ± 0.7 | 44.4 ± 1.2 | 55.8 ± 0.3 | 86.0 ± 0.6 | 68.3 |
| WSCrossWeigh | 90.8 ± 0.7 | 42.0 ± 0.0 | 46.0 ± 0.4 | 84.0 ± 0.9 | 65.7 |
| WSCleanlab $_{lfs}$ | 87.2 ± 0.4 | 44.4 ± 0.7 | 58.9 ± 0.3 | 86.6 ± 0.3 | 69.3 |
| WSCleanlab $_{sgn}$ | 88.3 ± 0.5 | 44.6 ± 0.9 | 56.4 ± 0.3 | 85.1 ± 0.3 | 68.6 |
| **ULF** $_{rndm}$ | 92.8 ± 0.1 | 44.8 ± 0.5 | 58.0 ± 0.2 | 85.7 ± 0.5 | 70.3 |
| **ULF** $_{lfs}$ | 90.8 ± 0.9 | 44.0 ± 0.9 | 55.5 ± 0.4 | 70.0 ± 0.4 | 65.1 |
| **ULF** $_{sgn}$ | **94.6** ± 0.2 | **49.8** ± 1.0 | 58.2 ± 0.2 | **88.6** ± 0.3 | **72.8** |

Table 6: Results of the feature-based ULF compared towards the feature-based baselines. All results are averaged over 10 trials and reported with the standard error of the mean. The results marked with * are taken directly from Zhang et al. (2021).

details. The feature-based ULF and the corresponding baselines are realized in our experiments with a logistic regression model; the training data are encoded with TF-IDF vectors. The results of feature-based experiments are provided in Table 6.

**Results.** Our weakly supervised extensions to CrossWeigh and Cleanlab methods consistently outperform the base versions and most of other baselines, supporting our hypothesis of LFs' importance in applying cross-validation techniques to weakly supervised settings. Feature-based ULF also shows the best result overall on most datasets and even outperforms the model trained on YouTube data with manual annotations.

We also use feature-based ULF to compare the data splitting strategies. ULF$_{rndm}$ incorporates a standard cross-validation with random data splitting, disregarding any signal originating from the weak annotation. This approach can be viewed for estimating the ULF performance alone, independent of any weak signals. The lower performance performance of ULF$_{lfs}$ compared to other configurations may be due to multiple LF matches in many data samples, leading to multiple overlaps during cross-validation training (i.e., the samples were reestimated multiple times). In ULF$_{sign}$, on the contrary, each data sample is considered only once in each denoising round resulting in a better performance. However, even in the worst-performing settings (ULF$_{lfs}$ and ULF$_{rndm}$) our feature-based ULF outperforms the majority of the baselines.

The signature-based splitting, which demonstrated the best performance for feature-based ULF, was chosen for the fine-tuning-based ULF experiments (see Table 2).

## F Implementation Details

All our experiments used the validation set for hyper-parameter tuning, early stopping, and model selection. The gradient-based optimization was performed with AdamW Optimizer and linear learning rate scheduler. All results are reproducible with the seed value 1111.

ULF-specific parameter search space was defined heuristically. All parameter search spaces are provided in Table 7. The number of iterations $I$ was also estimated based on the validation set: initially, it is set to $I = 20$, but if training labels do not change after three iterations, the algorithm stops, and the last saved model is used for final testing. The actual number of iterations, alongside other hyper-parameter values, can be found in Tables 8 and 9. These tables show that a single iteration yields the optimal result in most scenarios, with two iterations being the second most commonly selected option.

In fine-tuning-based ULF, in addition to other hyper-parameters, we include the *label prediction* parameter: if it equals "soft", the probabilistic labels are used for training; otherwise ("hard") a label is the one-hot encoding of the most probable class (the ties are broken randomly).

In order to reduce computational load, we performed the random parameter search instead of the

grid search in all our experiments. Specifically, we tried 10 random parameter combinations from search space and selected the one which performed the best on the validation set. For feature-based ULF and corresponding baselines, the model with the retrieved best hyper-parameter values was run ten times with different initializations; the average values with the standard error of the mean are reported. For fine-tuning-based ULF and corresponding baselines, the model with the retrieved best hyper-parameter values was run once on the test set; this value is reported. The retrieved hyper-parameters are provided in Table 8 for the best feature-based ULF setting, $\text{ULF}_{sng}$, and in Table 9 for all fine-tuning and cosine combinations of the fine-tuning-based ULF.

Both feature-based and fine-tuning-based ULF were implemented with Python and PyTorch (Paszke et al., 2019) in the setting of the weak supervision framework *Knodle* (Sedova et al., 2021). By providing access to all WS components Knodle allowed us to implement and benchmark all algorithms described above. The pre-trained language models were downloaded from HuggingFace (https://huggingface.co/models). We followed the Wrench (Zhang et al., 2021) encoding method and used their implementation for most of the baselines (apart from Cleanlab and CrossWeigh which are not included in the Wrench framework).

Feature-based ULF experiments were performed on a machine with a CPU frequency of 2.2GHz with 40 cores. Fine-tuning-based ULF experiments were run on a single Tesla V100 GPU on Nvidia DGX-1. The full setup took 20 hours on average for each dataset for feature-based settings and 96 hours for fine-tuning-based settings.

| Hyperparameter | Values |
|---|---|
| *Feature-based ULF* | |
| Multiplying coefficient $p$ | 0.1, 0.2, 0.3, 0.4, 0.5, 0.7, 0.9 |
| Learning rate $lr$ | 1e-1, 1e-2, 1e-3, 1e-4 |
| Number of folds $k$ | 3, 5, 8, 10, 15, 20 (w.r.t. #LFs) |
| Number of iterations $I$ | 1, 2, 3, 4, 5, 10 |
| Non-labeled data rate | 0, 0.5, 1, 2, 3 |
| *Fine-tuning-based ULF* | |
| Multiplying coefficient $p$ | 0.1, 0.3, 0.5, 0.7, 0.9 |
| Learning rate $lr$ | 1e-4, 1e-5, 1e-6 |
| Number of folds $k$ | 3, 5, 7 (w.r.t. #LFs) |
| Number of iterations $I$ | 1, 2, 3, 4 |
| Confident regular. weight $\lambda$ | 0.01, 0.1 |
| The confident threshold $\xi$ | 0.2, 0.4, 0.6, 0.8 |
| Label prediction | soft, hard |

Table 7: Hyperparameter values tried in a grid search in feature-based ULF and fine-tuning-based ULF.

|  | YouTube | Spouse | TREC | SMS |
|---|---|---|---|---|
| Multiplying coefficient $p$ | 0.5 | 0.2 | 0.3 | 0.1 |
| Learning rate $lr$ | 1e-2 | 1e-2 | 1e-1 | 1e-1 |
| Number of folds $k$ | 8 | 3 | 3 | 10 |
| Number of iterations I | 5 | 1 | 1 | 2 |
| Non-labeled data rate $\lambda$ | 0 | 3 | 1 | 0.5 |

Table 8: Feature-based ULF$_{sng}$ selected hyperparameters.

|  | YouTube | Spouse | TREC | SMS | Yorùbá | Hausa |
|---|---|---|---|---|---|---|
| *Fine-tuning-based ULF$_{FT\_FT}$ selected hyperparameters.* | | | | | | |
| Multiplying coefficient $p$ | 0.5 | 0.1 | 0.1 | 0.3 | 0.1 | 0.1 |
| Learning rate $lr$ | 1e-4 | 1e-06 | 1e-05 | 1e-05 | 1e-06 | 1e-06 |
| Number of folds $k$ | 2 | 3 | 3 | 3 | 2 | 2 |
| Number of iterations I | 1 | 2 | 1 | 1 | 2 | 2 |
| Confident regularization weight $\lambda$ | 0.1 | 0.01 | 0.1 | 0.1 | 0.1 | 0.05 |
| Confident threshold $\xi$ | 0.8 | 0.2 | 0.6 | 0.8 | 0.2 | 0.8 |
| Label prediction | soft | soft | soft | soft | soft | soft |
| *Fine-tuning-based ULF$_{FT\_COS}$ selected hyperparameters.* | | | | | | |
| Multiplying coefficient $p$ | 0.5 | 0.1 | 0.3 | 0.3 | 0.1 | 0.1 |
| Learning rate $lr$ | 1e-4 | 1e-06 | 1e-06 | 1e-05 | 1e-05 | 1e-05 |
| Number of folds $k$ | 2 | 3 | 5 | 3 | 2 | 2 |
| Number of iterations I | 1 | 3 | 1 | 1 | 2 | 1 |
| Confident regularization weight $\lambda$ | 0.1 | 0.01 | 0.01 | 0.1 | 0.1 | 0.05 |
| Confident threshold $\xi$ | 0.8 | 0.2 | 0.8 | 0.8 | 0.2 | 0.4 |
| Label prediction | soft | hard | soft | soft | soft | soft |
| *Fine-tuning-based ULF$_{COS\_FT}$ selected hyperparameters.* | | | | | | |
| Multiplying coefficient $p$ | 0.1 | 0.1 | 0.1 | 0.3 | 0.1 | 0.1 |
| Learning rate $lr$ | 1e-06 | 1e-06 | 1e-05 | 1e-06 | 1e-06 | 1e-05 |
| Number of folds $k$ | 3 | 3 | 3 | 5 | 7 | 2 |
| Number of iterations I | 2 | 1 | 1 | 2 | 4 | 1 |
| Confident regularization weight $\lambda$ | 0.1 | 0.01 | 0.1 | 0.01 | 0.05 | 0.05 |
| Confident threshold $\xi$ | 0.2 | 0.2 | 0.6 | 0.4 | 0.4 | 0.4 |
| Label prediction | soft | hard | soft | soft | soft | soft |
| *Fine-tuning-based ULF$_{COS\_COS}$ selected hyperparameters.* | | | | | | |
| Multiplying coefficient $p$ | 0.7 | 0.1 | 0.3 | 0.1 | 0.1 | 0.1 |
| Learning rate $lr$ | 1e-05 | 1e-06 | 1e-06 | 1e-05 | 1e-06 | 1e-05 |
| Number of folds $k$ | 5 | 3 | 5 | 7 | 7 | 2 |
| Number of iterations I | 4 | 2 | 2 | 1 | 1 | 2 |
| Confident regularization weight $\lambda$ | 0.01 | 0.01 | 0.01 | 0.05 | 0.05 | 0.05 |
| Confident threshold $\xi$ | 0.2 | 0.2 | 0.8 | 0.4 | 0.4 | 0.4 |
| Label prediction | soft | hard | soft | hard | soft | hard |

Table 9: Fine-tuning-based ULF selected hyperparameters.