# OpenReview forum: "ULF: Unsupervised Labeling Function Correction using Cross-Validation for Weak Supervision"
_EMNLP/2023/Conference — EMNLP 2023 Main_

### Official Review · Reviewer_PGPZ · 2023-07-30

**Typos Grammar Style And Presentation Improvements:** overall is ok, easy to read and follow.
**Soundness:** 3

**Excitement:**

4: Strong: This paper deepens the understanding of some phenomenon or lowers the barriers to an existing research direction.

**Missing References:**

for cosine training, any reference?

**Paper Topic And Main Contributions:**

This paper is about modeling soft labeling function (or label function denoising) for a weakly supervised setting, i.e., to learn/update the label functions using k-fold cross validation. They normally started with hard label functions like keyword-based, or regular expression-based or heuristics).  The assumption is the noise specific to some label functions (e.g., keywords, regular-expression-based) can be detected by training a model that does not use those label functions signals and then comparing its predictions to the labels generated by the held-out label functions.
They proposed an interesting new method, i.e., the first to adapt cross-validation denoising methods to WS problems and refine the LFs to class allocation in the WS setting.  I appreciate the authors' efforts in providing detailed and well-document appendix to ensure the reproducibility of their work

**Questions For The Authors:**

author mentioned cosine training on page 4, but not explanation or reference given.

Author replied to this.

**Reasons To Accept:**

Although with limitations,  overall they have presented a study with clear and comprehensive explanation of their methodology and experimental setup. The authors have provided a thorough comparison of their approach with baselines and existing methods, which adds value to the paper and strengthens its contributions.  i think this work contributes to the field with novel findings that are likely to be of interest to the scientific community.

**Reasons To Reject:**

no

**Reproducibility:**

4: Could mostly reproduce the results, but there may be some variation because of sample variance or minor variations in their interpretation of the protocol or method.

**Reviewer Confidence:**

3: Pretty sure, but there's a chance I missed something. Although I have a good feel for this area in general, I did not carefully check the paper's details, e.g., the math, experimental design, or novelty.

---

> ### Author Rebuttal · Authors · 2023-08-28
>
> Thank you for your positive review!
>
> We do not reference the Cosine paper on page 4 because we have already referenced it on page 3, line 199. The explanation of the Cosine training setup is provided in Appendix B, Algorithm 3. We will consider moving it to the main paper when we have an additional page.

---

### Official Review · Reviewer_jCVR · 2023-08-06

**Soundness:** 3

**Excitement:**

4: Strong: This paper deepens the understanding of some phenomenon or lowers the barriers to an existing research direction.

**Paper Topic And Main Contributions:**

This work proposes a novel approach to enhancing the quality of weak labels using k-fold cross-validation. Specifically, the authors introduce a new method called ULF (Unsupervised Labeling Function correction) that successfully labels samples with no LFs matched. The authors conduct extensive experiments using feature-based and pre-trained models to demonstrate the effectiveness of their method.

**Reasons To Accept:**

- The paper proposes a novel approach to enhancing the quality of weak labels using k-fold cross-validation.
- They introduce a method called Unsupervised Labeling Function(ULF) correction that successfully labels samples with no labeling functions matched.
- The work clearly outlines the motivation for re-estimate the joint distribution between labeling functions and class labels and carry out experiments evaluating the same.
- The authors conduct extensive experiments using feature-based and pre-trained models to demonstrate the effectiveness of their method
- The results show that the proposed solution performs comparably or better than the current solutions/baselines.
- Overall, the paper is well-written, and the authors provide a clear and concise explanation of their approach and results.

**Reasons To Reject:**

- The work does not have any ablation experiments to understand the impact of different types of labeling functions
- The work would benefit from a discussion section that analyzes the results.

**Reproducibility:**

3: Could reproduce the results with some difficulty. The settings of parameters are underspecified or subjectively determined; the training/evaluation data are not widely available.

**Reviewer Confidence:**

3: Pretty sure, but there's a chance I missed something. Although I have a good feel for this area in general, I did not carefully check the paper's details, e.g., the math, experimental design, or novelty.

---

> ### Author Rebuttal · Authors · 2023-08-28
>
> Thank you for your positive review!
>
> The results of the ablation experiments (i.e., the experiments with different combinations of cross-validation and final models) are provided in Table 4. For the best-performing setting, we also provide the information about the impact of each labeling function separately: see Figure 2 with the exact changes made by ULF (e.g., how each labeling function allocation was changed) and Table 3 with the recalculated final labels.
>
>
> To offer deeper understanding regarding the influence of labeling functions in a different scenario, we now provide the transformation of the T matrix in another setting (i.e., cross-validation is done with the Cosine training step):
>
> T matrix after the first ULF iteration:
>
> | HAM | SPAM |
> |------|------|
> | 0.05 | 0.95 |
> | 0.06 | 0.94 |
> | 0.02 | 0.98 |
> | 0.03 | 0.97 |
> | 0.98 | 0.02 |
> | 0.03 | 0.97 |
> | 0.99 | 0.01 |
> | 0.99 | 0.01 |
> | 0.98 | 0.02 |
> | 0.95 | 0.05 |
>
> T matrix after the second ULF iteration:
>
> | HAM | SPAM |
> |------|------|
> | 0.09 | 0.91|
> | 0.10 | 0.90|
> | 0.02 | 0.98|
> | 0.07 | 0.93|
> | 0.91 | 0.09|
> | 0.10 | 0.90|
> | 0.93 | 0.07|
> | 0.94 | 0.06|
> | 0.94 | 0.06|
> | 0.90 | 0.10|
>
> Here, same as in Figure 2, each row corresponds to one labeling function (see Appendix B for the examples of labeling functions for this and other datasets) and each column - to one class.
> Using a more powerful Cosine training makes the corrections to the T matrix much stronger and more intense; in our case, these stronger corrections end up causing more problems (compared to the simple fine-tuning) instead of improvements, as also demonstrated by the numbers in our experiments. Conversely, incorporating Cosine training into the end model training proves advantageous and enhances the final results. This analysis will be added in the camera-ready version.
>
> Also, please note that in order to ensure the reproducibility of our experiments, we report the settings of parameters (i.e., all hyperparameters search range and selected values for all experiments) in Appendix G, Tables 7-9.

---

### Official Review · Reviewer_D61f · 2023-08-10

**Typos Grammar Style And Presentation Improvements:** Not found.
**Soundness:** 3

**Excitement:**

2: Mediocre: This paper makes marginal contributions (vs non-contemporaneous work), so I would rather not see it in the conference.

**Missing References:**

1. Karamanolakis, Giannis, et al. "Self-training with weak supervision." arXiv preprint arXiv:2104.05514 (2021).
2. Maheshwari, Ayush, et al. "Semi-supervised data programming with subset selection." arXiv preprint arXiv:2008.09887 (2020).

**Paper Topic And Main Contributions:**

The topic of the paper is:
Unsupervised Labeling Function Correction using Cross-Validation for Weak Supervision.

The main contributions of the paper are:
The paper proposes a method to denoise the weakly supervised data by providing a kind of weighting mechanism for Labelling functions. It optimizes the assignment of labeling functions (LFs) to classes by recalculating this allocation using well-established cross-validated samples known for their high reliability. However, there are many label aggregation approaches (e.g. ASTRA, SPEAR, ImplyLoss) already available that perform a similar kind of task. Hence the contribution might be incremental.

**Questions For The Authors:**

See weaknesses.

**Reasons To Accept:**

Reasons to accept are:
1. The paper is well-described and written.
2. Results provided in the paper state that except one dataset, it has performed better than other approaches. However, there are many other methods (e.g. ASTRA, SPEAR, ImplyLoss) that should be added in comparison tables as these are the new state-of-arts.

**Reasons To Reject:**

Reasons to reject are:
1. In line 71, the authors only try majority voting as the label aggregation approach. However, many other label aggregation approaches, e.g. ASTRA, SPEAR, Imply Loss etc., choose the LFs sensibly in case of a tie by weighing the LFs based on previously correctly assigned labels. There is no comparison of the proposed approach with them as well.

2. Experimental section does not have any information about computational requirements, and hyperparameters used and it just contains the description of the dataset. The section on the case study could be made small and more details about hyperparameter choices and the computational requirements could be added.

3. Authors really need to manage the space as there are multiple redundant figures (e.g. figure 3 solely gives the interpretation of the changes in the label and figure 2 could be moved to the appendix or vice versa).

**Reproducibility:**

5: Could easily reproduce the results.

**Reviewer Confidence:**

2: Willing to defend my evaluation, but it is fairly likely that I missed some details, didn't understand some central points, or can't be sure about the novelty of the work.

---

> ### Author Rebuttal · Authors · 2023-08-28
>
> Thank you for your review!
>
> 1. In line 71, we describe the majority voting as the simplest label aggregation approach and discuss its downsides in order to motivate *our* label aggregation approach ULF. We use majority vote *only* as a baseline and compare our method to it and 5 other label aggregation methods: MeTaL, Snorkel-DP, FlyingSquid, WeaSEL, and the most recent FABLE. One cannot compare ULF to ASTRA (Karamanolakis et al., 2021), SPEAR (Maheshwari et al., 2020), or ImplyLoss (Awasthi et al., 2020) as these methods use and rely on manually labeled data; those are semi-supervised methods that leverage weakly annotated data in addition to manually labeled data. On the contrary, ULF does not make use of any manual supervision at all and is applicable in actual weakly supervised settings, where a classifier can only be trained *exclusively* on the automatically (weakly) labeled data. We also point it out in lines 805-808: *"...Note that ULF does not use manually annotated data and cannot be directly compared to models that do (Karamanolakis et al., 2021; Awasthi et al., 2020)..."*. Therefore, we compare ULF towards the label aggregation methods that also do not use manually labeled data in order to provide a fair comparison. We will make this more clear in the final version of the paper. Also note that the methods we compare ULF to are more recent: WeaSEL (Cachay et al., 2021) and the most recent SOTA method FABLE (Zhang et al., 2023).
> 2. We report all hyperparameters search range and selected values for all experiments, as well as other implementation details and computational requirements in Appendix G, Tables 7-9. We will consider using the additional page for moving this information to the main sections upon acceptance.
> 3. Thank you for your suggestion to move Figure 2 into the Appendix, we will consider it for the final version. Figure 3 in the Appendix gives an additional illustration of the type of changes induced by the algorithm - since space is not a critical factor in the Appendix, we decided to show it there as we hope it can aid the understanding of ULF.

---

### Meta-Review · Area_Chair_Qyar · 2023-09-19

**Recommendation:** 4

**Metareview:**

The paper introduces an approach for denoising weakly supervised data, particularly through the introduction of a weighting mechanism for labeling functions and the Unsupervised Labeling Function (ULF) correction. Based on the discussions during the rebuttal phase, below is the summary of the main aspects of the work.

Reasons to Accept:

Clarity and Presentation: The paper is well-described and well-written. The clear and organized presentation enhances the overall understanding of the research, making it accessible to a broad audience.

Competitive Performance: The paper presents compelling results, demonstrating that, with the exception of one dataset, the proposed approach consistently outperforms other existing methods. This highlights the effectiveness and competitiveness of the proposed method.

Motivated Research: The work is well-motivated and provides a clear rationale for the need to re-estimate the joint distribution between labeling functions and class labels. The experiments conducted effectively evaluate this motivation, strengthening the paper's credibility.

Extensive Experimentation: The authors have conducted a comprehensive set of experiments, including feature-based and pre-trained models, to demonstrate the effectiveness of their method. This thorough experimentation adds significant value to the research. The appendices provide extra analyses and results.

Comparative Performance: The results consistently indicate that the proposed solution either performs comparably or surpasses current solutions and baselines. This underscores the competitiveness and utility of the proposed approach.

Reasons to Reject:

Deeper Discussion Section Needed: While the paper excels in its description and presentation, it would significantly benefit from a more extensive discussion section. A deeper analysis of the results, including their implications and nuances, is essential to provide readers with a more comprehensive understanding of the research.

Future Directions and Next Steps: As a short paper, considering the inclusion of a section that outlines potential future research directions and next steps would be valuable. Identifying areas where the proposed approach could be extended, improved, or applied would enhance the paper's significance.

In conclusion, there is convergence on the reviews considering that the paper exhibits substantial promise with its clarity, strong presentation, innovative approach, motivated research, extensive experimentation, and competitive performance. To further elevate its contribution and overall quality, it is recommended to expand the discussion section and consider adding a section on future research directions.

---

### Decision · Program_Chairs · 2023-10-07

**Decision:**

Accept-Main

**Comment:**

The paper introduces an approach for denoising weakly supervised data, particularly through the introduction of a weighting mechanism for labeling functions and the Unsupervised Labeling Function (ULF) correction. Based on the discussions during the rebuttal phase, below is the summary of the main aspects of the work.

Reasons to Accept:

Clarity and Presentation: The paper is well-described and well-written. The clear and organized presentation enhances the overall understanding of the research, making it accessible to a broad audience.

Competitive Performance: The paper presents compelling results, demonstrating that, with the exception of one dataset, the proposed approach consistently outperforms other existing methods. This highlights the effectiveness and competitiveness of the proposed method.

Motivated Research: The work is well-motivated and provides a clear rationale for the need to re-estimate the joint distribution between labeling functions and class labels. The experiments conducted effectively evaluate this motivation, strengthening the paper's credibility.

Extensive Experimentation: The authors have conducted a comprehensive set of experiments, including feature-based and pre-trained models, to demonstrate the effectiveness of their method. This thorough experimentation adds significant value to the research. The appendices provide extra analyses and results.

Comparative Performance: The results consistently indicate that the proposed solution either performs comparably or surpasses current solutions and baselines. This underscores the competitiveness and utility of the proposed approach.

Reasons to Reject:

Deeper Discussion Section Needed: While the paper excels in its description and presentation, it would significantly benefit from a more extensive discussion section. A deeper analysis of the results, including their implications and nuances, is essential to provide readers with a more comprehensive understanding of the research.

Future Directions and Next Steps: As a short paper, considering the inclusion of a section that outlines potential future research directions and next steps would be valuable. Identifying areas where the proposed approach could be extended, improved, or applied would enhance the paper's significance.

In conclusion, there is convergence on the reviews considering that the paper exhibits substantial promise with its clarity, strong presentation, innovative approach, motivated research, extensive experimentation, and competitive performance. To further elevate its contribution and overall quality, it is recommended to expand the discussion section and consider adding a section on future research directions.